# Effects of Pressurized Aeration on the Biodegradation of Short-Chain Chlorinated Paraffins by *Escherichia coli* Strain 2

**DOI:** 10.3390/membranes12060634

**Published:** 2022-06-19

**Authors:** Yongxing Qian, Wanling Han, Fuhai Zhou, Bixiao Ji, Huining Zhang, Kefeng Zhang

**Affiliations:** 1School of Civil Engineering and Architecture, NingboTech University, Ningbo 315000, China; qianyx312@nbt.edu.cn (Y.Q.); hanwanling0325@outlook.com (W.H.); jibixiao@nbt.edu.cn (B.J.); kfzhang@nit.zju.edu.cn (K.Z.); 2Ningbo Research Institute, Zhejiang University, Ningbo 315100, China; 3Zhejiang Haiyi Environmental Protection Equipment Engineering Co., Ltd., Quzhou 324000, China; zjhbfw@126.com

**Keywords:** short-chain chlorinated paraffins, *Escherichia coli* strain 2, pressurized aeration, removal mechanism

## Abstract

Short-chain chlorinated paraffins (SCCPs) were defined as persistent organic pollutants in 2017, and they can migrate and transform in the environment, accumulate in organisms, and amplify through the food chain. Although they pose a serious threat to environmental safety and human health, there are few papers on their removal. The current SCCP removal methods are expensive, require severe operating conditions, involve time-consuming biological treatment, and have poor removal specificities. Therefore, it is important to seek efficient methods to remove SCCPs. In this paper, a pressurized reactor was introduced, and the removal performance of SCCPs by *Escherichia coli* strain 2 was investigated. The results indicated that moderate pure oxygen pressurization promoted bacterial growth, but when it exceeded 0.15 MPa, the bacterial growth was severely inhibited. When the concentration of SCCPs was 20 mg/L, the removal rate of SCCPs was 85.61% under 0.15 MPa pure oxygen pressurization for 7 days, which was 25% higher than at atmospheric pressure (68.83%). In contrast, the removal rate was only 69.28% under 0.15 MPa air pressure. As the pressure continued to increase, the removal rate of SCCPs decreased significantly. The total amount of extracellular polymeric substances (EPS) increased significantly upon increasing the pressure, and the amount of tightly bound EPS (TB-EPS) was higher than that of loosely bound EPS (LB-EPS). The pressure mainly promoted the secretion of proteins in LB-EPS. Furthermore, an appropriate pure oxygen pressure of 0.15 MPa improved the dehydrogenase activity. The gas chromatography–mass spectrometry (GC–MS) results indicated that the degradation pathway possibly involved the cleavage of the C–Cl bond in SCCPs, which produced Cl^−^, followed by C–C bond breaking. This process degraded long-chain alkanes into short-chain alkanes. Moreover, the main degradation products detected were 2,4-dimethylheptane (C_9_H_20_), 2,5-dimethylheptane (C_9_H_20_), and 3,3-dimethylhexane (C_8_H_18_).

## 1. Introduction

Chlorinated paraffins (CPs) are chlorinated from n-alkane raw materials, whose degree of chlorination is usually 30–70 wt.%. According to the different carbon chain lengths, CPs can be divided into short-chain chlorinated paraffins (SCCPs, C_10_–C_13_), middle-chain chlorinated paraffins (MCCPs, C_14_–C_17_), and long-chain chlorinated paraffins (LCCPs, C_18_–C_30_) [1]. Due to their excellent flame retardancy, electrical insulation, and low volatility, CPs can generally be used as high-temperature lubricants, plasticizers, flame retardants, adhesives, paints, rubbers, and sealants as additives [2,3]. China is a major producer and exporter of CPs [4], with a total production capacity of 2.08 million tons and an actual output that reached 0.835 million tons in 2018. Compared with MCCPs and LCCPs, SCCPs pose a greater environmental threat. They are carcinogenic, teratogenic, and mutagenic and have the highest toxicity. They were finally included in the list of persistent organic pollutants (POPs) in Annex A of the *Stockholm Convention on Persistent Organic Pollutants* in 2017 [5]. As one of the parties to the *Stockholm Convention* and a major producer and user of halogenated flame retardants, China faces tremendous pressure and is making every effort to abate SCCPs in the environment.

To date, SCCPs have been reduced by treatment with metal sodium dispersion dechlorination [6], zero-valent iron [7], and photocatalytic degradation [8,9,10]. However, the current use of physicochemical treatment methods to remove SCCPs is expensive, and the operating conditions are also severe. Biological methods are favored over physicochemical methods because of their simple process, low costs, and environmental friendliness. Among them, bacterial degradation methods show greater prospects for the treatment of SCCPs. Allpress et al. [11] first reported the Gram-positive strain of *Rhodococcus* sp. S45-1, which utilized SCCPs as their only carbon and energy source for metabolic degradation. However, the strain required a long degradation time of 30–100 days. Gram-negative bacteria can also degrade SCCPs. Heath et al. [12] screened *Pseudomonas* sp. strain 273, which dechlorinated chlorinated alkanes, but just like the strain screened by Allpress, the dechlorination cycle of this strain lasted 20 days or longer. Similarly, Lu [13] obtained *Pseudomonas* strain N35 from dehydrated sludge collected in a secondary sedimentation tank. The pure strain effectively degraded SCCPs with a dechlorination rate of 57.5% within 20 days using SCCPs as the carbon and energy source. Adding bacteria to the sludge removed 73.4% of SCCPs, but the degradation time was too long, requiring 30 days. Therefore, improving the biodegradation efficiency of pollutants is a crucial factor for expanding the applications of biological methods.

Dissolved oxygen (DO) can influence the efficiency of aerobic microbial wastewater treatment. According to Henry’s law and the double-membrane theory of gas transfer, increasing the gas pressure in a bioreactor will increase the concentration of saturated dissolved gases in water, breaking through the atmospheric supply limit, and thereby promoting the biodegradation of organics in wastewater. Pressurization technology has been widely used to improve the biological treatment of activated sludge and granular sludge [14,15,16,17]. Previous studies have shown that pressure affects the growth state, morphology, and secretion of microorganisms, but a few researchers have pointed out that moderate pressure will not cause microbial changes. Jin et al. [15] reported that moderate pressure helped the degradation of pesticide wastewater by activated sludge. Under 0.3 MPa, the removal rate of chemical oxygen demand (COD) in pesticide wastewater was much higher than that under atmospheric pressure. Zhang et al. [17] pointed out that the removal rate of COD was the highest when the pressure was 0.4 MPa. Due to an increase in the DO concentration in water under moderate oxygen pressure, DO was no longer the limiting factor for microbial degradation of pollutants.

Extracellular polymeric substances (EPS) are microbial secretions that are metabolites or autolysates that are attached to or surround microorganisms. They are mainly composed of proteins, polysaccharides, and a small number of nucleic acids. Various studies have shown that EPS play a pivotal role in pollutant removal processes. EPS can adsorb pollutants, such as dyes [18], refractory organics [19], and heavy metals [20], or transfer phosphorus and electrons to promote interactions between pollutants and microbes [21,22]. Moreover, EPS affect bacterial attachment and movement and is often the culprit for the fouling of membrane modules [23]. Shi et al. [24] provided an excellent review of the role of EPS in biological treatment methods. They showed that the characteristics of EPS were affected by the feed substances, operating conditions, and substances in the environment. Despite these studies, the impact of pressure on EPS has not been investigated. Thus, studying the impact of EPS under pressure on the microbial degradation of pollutants can supplement our understanding of the effect of EPS at atmospheric pressure and help to provide certain basic experimental data for adding membrane modules to the pressurized membrane bioreactor and preventing its fouling.

Here, to improve the efficiency of the microbial removal of SCCPs, pressurized bioreactors were utilized. In this work, the synergistic effect of *E. coli* strain 2 on the removal of SCCPs was studied in a pressurized system, and the removal performance under different working conditions (rotation speed, initial SCCP concentration, pressure, and type of gas) was compared and analyzed in pressurized bioreactors. Moreover, the transformation law of *E. coli* strain 2 parameters, including bacterial growth, EPS, and dehydrogenase (DHA), was revealed in the pressurized system. Moreover, the changes in EPS that were the culprit for the fouling of membrane modules in different pressurized reactors were investigated. Based on this, changes in bacterial morphology and possible degradation pathways of SCCPs in the pressurized system were clarified.

## 2. Materials and Methods

### 2.1. Reagents and Materials

Industrial short-chain chlorinated paraffins were purchased from a chlorinated paraffin production factory (Henan, China). Methanol (chromatographically pure), acetone, reagent sodium carbonate, sodium bicarbonate, and pure xylene were purchased from Sinopharm (Shanghai, China). All the other chemicals and reagents used were of analytical grade. Ultrapure water (18.2 MΩ·cm) was obtained from a Milli-Q purification system (Merck Millipore, Darmstadt, Germany). The SCCP-degrading pure bacterium *E. coli* strain 2 (preliminary experiments implied that the bacteria had an excellent dechlorination effect on SCCPs) used in this study was quickly screened from activated sludge in a local wastewater treatment plant (Ningbo, China).

### 2.2. The Pressurized Bioreactor

The pressurized bioreactor used in this work is seen in Appendix A. The outer diameter of the tank was 64 mm, and the height was 140 mm. The pressurized bioreactor was made of steel with a 48 mm diameter, 110 mm height, and 200 mL effective volume. All the pressurization experiments were conducted in the tank (Appendix A). The inlet pipe (inner pipe length: 90 mm) went deep into the bottom of the tank, which was conducive to gas–liquid two-phase mass transfer. This prolonged the residence time of reactive substances in the tank and reduced the impact of high-pressure airflow on the inner wall of the tank.

During the experiment, the reactor shell was immersed in a constant-temperature heating magnetic stirrer as a water bath, and the built-in rotor stirred the reaction medium to improve the probability of contact between microorganisms and pollutants. Furthermore, batch experiments were conducted within the operating pressure range of 0–0.3 MPa. All the pressures in this paper were the operating pressure, that is, the gauge pressure of the pressurized reactor (where 0 MPa pressure in the reactor represents 0.1 MPa atmospheric pressure). In addition, the experimental temperature of all the reactors in this study was set to 33 °C (Appendix A). The pressurized reactor was cleaned with acetone before the reaction and then sterilized by an ultraviolet lamp.

### 2.3. Preparation of SCCP Stock Solution and Culture Medium

Since SCCPs are difficult to dissolve in water, acetone was utilized as a cosolvent. SCCPs (0.1 g) were dissolved in 100 mL acetone to obtain a 1 g/L SCCPs stock solution, which was stored at 4 °C for further use.

The beef extract peptone medium was composed of 5.0 g/L beef extract, 10.0 g/L peptone, 5.0 g/L K_2_SO_4_, and 20.0 g/L agar (if solid culture). The mixture was autoclaved at 121 °C for 30 min. The pH was in the range of 7.0–7.4.

The SCCP degradation medium was obtained by adding a definite volume of SCCP stock solution (sterilized by a 0.22 μm membrane filter) to the autoclaved beef extract peptone medium.

### 2.4. Biodegradation Experiments

To improve the utilization rate of the SCCP carbon source by *E. coli* strain 2, the amount of additional carbon source was reduced. The concentration of beef extract in the nutrient solution was reduced to 0.2 g/L. The effect of rotation speed on the removal of SCCPs and bacterial growth was studied at 320, 640, and 960 rpm. The effect of different initial concentrations of SCCPs on the bacterial removal of SCCPs was studied at 20, 30, and 50 mg/L concentrations at 0.15 MPa pressure. Moreover, SCCP (20 mg/L) removal experiments were conducted under different types of pressurized gas (pure oxygen and air) and different pressures (0–0.3 MPa). In the early stages of the first 2 days, samples of effluent water were collected every 12 h. Afterward, samples were taken every other day to determine bacterial growth and SCCP concentration, and the EPS and DHA were measured after 7 days.

For the biodegradation experiments, exponentially growing cells were harvested by centrifugation (9000× *g*, 5 min) and washed three times with 0.01 M phosphate buffer (pH 7.0) before use.

### 2.5. Analytical Methods

#### 2.5.1. Determination of Bacterial Indices

The measurement of bacterial indices in the experiments included the bacterial biomass (expressed by the bacteria’s optical density (OD_600_)), volatile suspended solids (VSS), EPS (mainly loosely bound EPS (LB-EPS) and tightly bound EPS (TB-EPS]), and DHA. A stable reaction was achieved on the 7th day of culture. The bacterial growth was measured by a UV–VIS spectrophotometer to measure the OD_600_, and the determination of DHA was performed as described by Lee et al. [25].

#### 2.5.2. SCCPs, EPS, and CSH Determination

The concentration of SCCPs was determined by gas chromatography (GC-2030, Shimadzu, Kyoto, Japan) equipped with an electron capture detector with liquid–liquid extraction pretreatment. The specific method was as described by Jiang’s research group [26] with modifications. The specific determination conditions were as follows: a 1 μL sample was injected by an AOC-20i automatic injector into an InertCap 5 capillary column (30 m × 0.25 mm × 0.25 μm, Shimadzu). The injection inlet temperature was 300 °C. Samples were injected in split mode with a split ratio of 30:1 after pretreatment. Nitrogen was the carrier gas at a constant flow rate of 1.1 mL/min. The temperature of the detector (ECD) was 320 °C, and the oven temperature program for the SCCPs was as follows: hold at 100 °C for 1 min, increase to 160 °C at 30 °C/min, hold for 5 min, increase to 310 °C at 30 °C/min, hold for 17 min.

LB-EPS and TB-EPS were extracted using a modified thermal extraction method [27]. Proteins (PN) and polysaccharides (PS) were determined by Coomassie bright blue G-250 and phenol–sulfuric acid colorimetry [28]. This work used the Tromans thermodynamic equation [29] to describe the initial DO concentration inside the reactor under different pressures. To understand the cell surface hydrophobicity (CSH) of *E. coli* strain 2 before and after EPS extraction, the improved bath method [30] was used for determination. Detailed descriptions of Tromans thermodynamic equation and CSH value calculated Equation are provided in the supporting information.

#### 2.5.3. SEM and Metabolite Identification

The samples obtained during the 7-day pressurization reaction were analyzed by scanning electron microscopy (SEM) (Zeiss Sigma 300, Jena, Germany) to observe the morphological changes of the bacterial surface structure and the EPS produced by the bacteria under different pressures.

The concentration of chloride ions was determined using an ion chromatograph (ICS-600, Dionex; Thermo Fisher Scientific, Waltham, MA, USA), equipped with a 250 mm IonPac AS23 column (inner diameter: 4 × 250 mm), as well as a guard column and a current suppressant. The eluant was 4.5 mm Na_2_CO_3_/0.8 mm NaHCO_3_. The flow rate was 1.0 mL/min, the column temperature was 30 °C, the suppression current was 25 mA, and the injection volume was 10 μL with nitrogen as the carrier gas.

To further clarify the removal of the metabolites of SCCPs by *E. coli* strain 2 under atmospheric pressure (0 MPa) and at optimal conditions (0.15 MPa) after reacting for 7 days, a 7890A/5975C gas chromatography–mass spectrometer (GC–MS, Agilent Technologies, Santa Clara, CA, USA) was employed. The intermediate products produced during the degradation of SCCPs were detected and identified.

### 2.6. Quality Assurance and Quality Control (QA/QC)

To prevent potential background contamination of SCCPs, all the bottles used in the sample extraction were either triple-rinsed with acetone or heated overnight at 500 °C in a muffle furnace. After that, all bottles were ultrasonically cleaned with ultrapure water three times before use. All the experiments were conducted in triplicate.

## 3. Results and Discussion

### 3.1. Effect of Rotation Speed on SCCP Removal

To explore the influence of different reaction speeds on SCCP removal by *E. coli* strain 2 under pressurization, 1% (*v*/*v*) *E. coli* strain 2 was added into the pressurized reactors (Appendix A) with 20 mg/L SCCPs. The pressure in the reactor was adjusted to 0.1 MPa using high-purity oxygen. The reaction speed in the pressurized reactor was adjusted by using magnetic stirring, which was set to a low speed of 320 rpm (the visible bacterial suspension was not sufficiently mixed, and the vortex was small), medium speed of 640 rpm (the visible bacterial suspension was just mixed, and the vortex was moderate), and high speed of 960 rpm (the visible bacterial suspension was just mixed, and the vortex was large). After 7 days, the OD_600_ of these bacteria was roughly stable at approximately 2.0 (Appendix A). The three growth curves showed good overlap, indicating that the effect of rotation speed on bacterial growth was insignificant under pressure. The removal efficiency of SCCPs at 320, 640, and 960 rpm were 77.81%, 77.42%, and 78.37% on day 7, respectively (Appendix A). In any case, a high rotation speed would cause wear to the rotor, and the mixing of the bacterial suspension was not adequate at a low rotation speed, while the wear to the rotor was small at a medium rotation speed, and the bacterial suspension was well blended. Therefore, 640 rpm was chosen for subsequent experiments.

### 3.2. Effect of Initial Concentration on Bacterial SCCP Removal

The growth of *E. coli* strain 2 decreased significantly upon increasing the SCCPs’ initial concentration under 0.15 MPa (Appendix A). After 48 h, the growth of *E. coli* strain 2 at each concentration stabilized. When the concentration of SCCPs increased from 20 to 50 mg/L, the OD_600_ of *E. coli* strain 2 decreased from 2.430 to 0.750 on the 7th day, indicating that a high concentration of SCCPs significantly inhibited the growth of *E. coli* strain 2. Generally, the concentration of pollutants affects the growth of microorganisms. For example, when the concentration of lindane was increased from 1 to 5 and 10 mg/L, by measuring the diameter of the colony in the petri dish, increasing the concentration of lindane partially or completely inhibited the mycelial growth of seven types of white-rot fungi [31]. A high concentration of tetracycline [32] (concentration > 100 mg/L) also delayed the growth of bacteria in the anaerobic digestion tank of a sewage treatment plant. When the concentration reached 250 mg/L, almost no bacterial growth was observed within 2 days. Furthermore, the abundance of filamentous bacteria in the aerobic biofilm reactor decreased upon increasing the antibiotic concentration from 5 to 25 mg/L, which confirmed that the pollutants were toxic to microorganisms and decreased their growth. Higher concentrations of pollutants were more toxic to the microorganisms. They significantly inhibited the growth and abundance of microorganisms [33]. Appendix A shows that the growth of bacteria was the best when the concentration was 20 mg/L. There was no significant difference between the growth of bacteria at atmospheric pressure and 0.15 MPa at this concentration, suggesting that moderate pressure did not affect the normal growth of bacteria.

Appendix A shows that the initial concentration had a significant impact on the SCCP removal within a certain concentration range. The SCCP removal rate reached the highest at 20 mg/L at 82.48% (0.15 MPa) and 71.64% (0 MPa) after 7 days. The removal rate of SCCPs at 0.15 MPa at the same concentration was better than that at atmospheric pressure. When the initial SCCP concentration was increased to 30 and 50 mg/L, respectively, the SCCP removal rate at 0.15 MPa decreased from 82.48% to 68.05% and 60.66%, respectively. The rate of decrease at 50 mg/L was significantly greater than that at 30 mg/L, implying that a higher concentration inhibited the removal of SCCPs by bacteria. As previously reported by Lu [13], the SCCP concentration decreased from 20, 50, 100, and 200 mg/L to 1.5, 10.3, 37.5, and 96.8 mg/L after 20 days of degradation, respectively. When studying the degradation of SCCPs by *Pseudomonas* N35, the corresponding degradation rates were 92.7%, 79.3%, 62.5%, and 51.6%, respectively. This is similar to the previous result in which the initial concentration of sulfamethoxazole was 5, 10, and 20 mg/L in the ultrasonic/PW12/KI/H_2_O_2_ combined system. The removal rates after 60 min were 95%, 94%, and 78%, respectively, demonstrating that the pollutant removal decreased significantly upon increasing the initial pollutant concentration [34]. Overall, a higher initial concentration of SCCPs significantly inhibited the removal of pollutants by pure strains.

### 3.3. Effect of Pressurization on Bacterial SCCP Removal

To further analyze the effect of pressure on the removal of SCCPs by *E. coli* strain 2, the effect of different pressures on the degradation of 20 mg/L SCCPs by *E. coli* strain 2 was studied at a rotating speed of 640 rpm. To explore the effect of *E. coli* strain 2 on SCCP removal during pressurization, 1% (*v*/*v*) of activated *E. coli* strain 2 in an atmospheric pressure shaker was sequentially transferred to a pressurized reactor with a pressure of 0–0.3 MPa. The SCCP removal rate, OD_600_, EPS, and DHA were measured when the reaction was stable. Thus, the influence of different pressures on the removal of SCCPs by the bacteria could be explored using different pressurization gases (oxygen or air).

#### 3.3.1. Pure Oxygen Pressurization

(1)Effects on bacterial growth and SCCP removal

Oxygen pressurization had a greater impact on the growth of *E. coli* strain 2. Upon increasing the pressure, the bacterial growth increased first and then decreased significantly. At pressures of 0, 0.05, 0.1, 0.15, 0.2, 0.25, and 0.3 MPa, the OD_600_ value of bacteria was 1.879, 1.928, 1.973, 2.077, 0.900, 0.775, and 0.479 on day 7, respectively. Particularly at 0.15 MPa, the OD_600_ value of the bacteria increased to 1.262 after 12 h of growth and reached 1.998 after 2 days (Figure 1a). The results showed that a moderate pressure (0.05–0.15 MPa) promoted the effect of *E. coli* strain 2, possibly because a higher pressure rapidly increased the dissolution rate of oxygen in water, and supersaturated DO promoted the growth of bacteria. However, when the pressure was 0.2, 0.25, and 0.3 MPa, the growth rate of *E. coli* strain 2 slowed, which was consistent with the results of Scoma et al., who found that a continuous culture of *Alcanivorax* sp. for 4 days at a pressure of 10 MPa (20 °C) significantly decreased in growth [35]. Moreover, Wang et al. [36] found that the optimal growth condition of Gram-positive pressure-tolerant *Bacillus* DSK25 isolated from sediments of the Japan Trench was 0.1 MPa. Its growth rate decreased upon increasing the pure oxygen pressure from 0.1 to 60 MPa.

The results of bacterial removal SCCPs (20 mg/L) under a high pure oxygen pressure are shown in Figure 1b. When the pressure increased from 0 to 0.15 MPa, the SCCP removal rate of the three reactors increased from 68.83% to 85.61% after 7 days, respectively, indicating that a moderate pressure promoted the removal of SCCPs. Under 0.15 MPa, the SCCP removal rate was nearly 25% higher than that of 0 MPa. A moderate increase in pressure accelerated the speed at which the DO passed through the bacterial cell membrane and accelerated the catabolism and anabolism of substances. Such a pressure maintained a high DO concentration in the culture medium, which improved the activity of microorganisms, thus accelerating the substrate utilization [37]. When the pressure increased from 0.15 to 0.3 MPa, the SCCP removal rate decreased from 85.61% to 60.33%. When the pressure exceeded 0.15 MPa, the DO was no longer a limiting factor for biochemical reactions and actually became an inhibiting factor. Excessively high pressure did not promote the normal growth of bacteria and may have even deformed and ruptured bacterial cells, ultimately killing them. Thus, it was not conducive to the degradation of pollutants by bacteria [38]. When Jin et al. [15] applied the pressurized activated sludge method to treat high-concentration pesticide wastewater, they found that when the critical pressure (0.3 MPa) was reached, the COD removal rate stopped increasing. Zhang et al. [17] showed that when the pressure further increased to 0.5 MPa, the COD removal rate declined from 86% to 81%. However, in this study, pressures above 0.15 MPa were not conducive to the bacterial degradation of SCCPs, and the pressure upper limit was much lower than that reported in the above literature. This might be because the pressure resistance of the screened pure bacteria was lower than that of the bacteria mixture in activated sludges.

(2)Variations in EPS

Many studies have shown that the microenvironment can affect the microbial secretion of EPS, which plays a key role in the removal of pollutants [18,19,20]. However, there are few studies on the changes of bacterial EPS under pressurization, and an investigation of the effect of oxygen pressure on EPS can supplement the results obtained under atmospheric pressure. Figure 2a displays the changes of EPS components (LB-EPS and TB-EPS) under high-purity oxygen pressurization. Upon increasing the pressure (0–0.3 MPa), the total EPS content increased significantly. When the pressure increased from 0 to 0.3 MPa, the EPS concentration increased from 53.06 to 171.38 mg/g VSS. TB-EPS occupied the dominant position, accounting for >50% of the total EPS. However, the growth rate of LB-EPS was faster and increased more rapidly compared with that of TB-EPS at pressures above 0.15 MPa (Figure 2a).

Moreover, upon increasing the pressure (Figure 2b), the protein (PN) in EPS increased by 14.80%, 59.35%, 101.04%, 486.59%, 605.65%, and 727.88%, indicating that extreme conditions, such as a high pressure, may significantly increase the PN produced by bacteria. It was noted that although the increased PS was not as obvious as that of PN upon increasing the pressure, the total content was still greater than that of PN. Figure 2c indicates the changes in the PS and PN in LB-EPS and TB-EPS under pressurization. Compared with TB-EPS, the pressure had a greater impact on the PN and PS in LB-EPS. For instance, the PN concentration in LB-EPS increased from 2.41 to 39.29 mg/g VSS at 0.3 MPa, an increase of approximately 15-fold. This meant that a high pressure promoted the secretion of PN in LB-EPS. Furthermore, Figure 2d shows that PN/PS generally increased upon increasing the pressure, which implied that PN became the dominant component of EPS when the pressure increased. However, the phenomenon in which EPS increased with pressure in this experiment was different from that reported by Zhuang et al. They reported that in a conventional oxygen-aerated membrane bioreactor, a higher DO concentration enhanced the metabolic activities of microorganisms in activated sludge, resulting in less EPS release or more EPS biodegradation [39]. Xu et al. [37] also found that under 0.3 MPa, more consumption of oxygen promoted the biodegradation of organic matter and consumed more EPS in activated sludge. This difference might be due to microorganisms because activated sludges (a mixture of bacteria) usually contain bacteria that degrade EPS, which may degrade EPS in the presence of sufficient oxygen. However, for pure bacteria, due to the lack of other bacteria that could degrade EPS [40], EPS increased even though there was sufficient oxygen.

(3)Variations in DHA

DHA is an intracellular enzyme that can degrade pollutants and directly reflect the degradative ability of organisms [41]. As plotted in Appendix A, the DHA increased from 0.391 to 0.477 mmol/g VSS when the pressure increased from 0 to 0.15 MPa. However, when the pressure was above 0.15 MPa, the DHA concentration decreased from 0.477 to 0.351 mmol/g VSS (0.3 MPa). This indicated that when the pressure exceeded the maximum pressure that *E. coli* strain 2 could withstand, the bacterial metabolism was significantly inhibited. It was hypothesized that the enhanced removal of SCCPs was attributed to the fact that pure oxygen provided a high DO concentration, which affected the biomass activity by affecting the enzyme activity. Since a higher pure oxygen transfer efficiency improved the biomass activity [42], pure oxygen met the high oxygen demand, even at lower flow rates [43]. However, when the oxygen transfer efficiency reached a high-enough level, the effect of oxygen transfer rate was not significant. DHA played an important role in the removal of SCCPs, and pure oxygen enhanced DHA, thus promoting the removal of pollutants.

#### 3.3.2. Air Pressurization

(1)Effects on bacterial growth and SCCP removal

Figure 3a shows the growth curve of bacteria under air pressurization. *E. coli* strain 2 grew well under 0 MPa, and the OD_600_ value was 1.879 on the 7th day. However, the OD_600_ value dropped significantly upon a further increase in air pressure, indicating that air pressurization greatly inhibited the growth of bacteria.

As shown in Figure 3b, the 20 mg/L SCCP removal rate did not greatly change under oxygen pressure. The SCCP removal rate increased and reached the highest value of 69.28% at 0.15 MPa at 7 days, but it was not much different from the removal rate at 0 MPa (68.83%). However, when the pressure continued to increase to 0.3 MPa, the SCCP removal rate decreased to 54.13%. This phenomenon may be caused by the adaptability of microorganisms to moderate pressure so that the SCCP removal rate remained stable [15,17]. In addition, since the oxygen content in the air was only 21% (*v*/*v*), and there was a low DO concentration during air pressurization, other reactors had a poor SCCP removal rate (<60%) at 0.05–0.3 MPa, except for 0.15 MPa. Therefore, it could be hypothesized that a DO deficiency weakened the promoting performance of pressure on SCCP removal, possibly because a high pressure and low DO were unsuitable microenvironments for bacterial survival. Thus, the inhibitory effect of pressure on SCCP removal increased under a low DO concentration.

(2)Variations in EPS

As shown in Figure 4a, the EPS first increased from 53.06 to 75.23 mg/g VSS, then decreased to 63.39 mg/g VSS as the pressure increased from 0 to 0.3 MPa. As the pressure increased, the EPS content tended to increase. Bacteria secrete more EPS to form a protective barrier in adverse environments, which promotes their survival [24]. TB-EPS accounted for more than 50% of the total EPS in all the reactors, accounting for 78.91%, 71.66%, 59.83%, 67.11%, 74.76%, 52.60%, and 58.17%. Figure 4b provides the concentrations of PN and PS produced by the bacteria, where PS accounted for the majority, but the content of PN increased with the pressure. The microbes secreted more extracellular PN that were resistant to adverse environments. In addition, the PS production also increased as part of the stress response [44]. Previous studies confirmed that PS also played a key role in microbial aggregation [45]. The EPS concentration reflects the DO content, and it has been reported that PS plays the dominant role in EPS produced by bacteria, regardless of the DO concentration [46]. Figure 4c shows that more PS was produced than PN under each pressure for both LB-EPS and TB-EPS. PS is generally more biodegradable than PN, which allows PN to more easily attach to the thallus and become part of the EPS produced, resulting in a higher PN/PS ratio in EPS, as shown in Figure 4d. Since the PS content in the total EPS was significantly higher than the PN content, a higher PS content decreased the PN/PS ratio, which indicated that bacteria tended to produce PS to adapt to a high-pressure environment. The low PN/PS ratio under 0.15 MPa in Figure 4d may have been because EPS was consumed to maintain microbial growth as the substrate was consumed. Wang et al. [47] reported the biodegradability of EPS and found that 50% EPS was utilized by its producers under aerobic starvation.

(3)Variations in DHA

Appendix A describes changes in the DHA under air pressurization. When the pressure was 0.05, 0.1, 0.15, 0.2, 0.25, and 0.3 MPa, the DHA was 0.153, 0.233, 0.371, 0.266, 0.239, and 0.224 mmol/g VSS, respectively. There was no significant positive correlation between the DHA and air pressure, but the highest DHA was obtained at 0.15 MPa. The reason was hypothesized that high-pressure conditions negatively impacted the cell structure of bacteria and their metabolic processes and viability [48]. Microorganisms can be adversely affected by high-pressure conditions, depending on the intensity of the pressure [49]. However, the SCCP removal rate significantly correlated with the DHA in this experiment. In addition, the DHA (0.391 mmol/g VSS) under 0 MPa was higher than that under pressurization, and the DHA decreased upon increasing the air pressurization time, which may be because the microorganisms exceeded the tolerable concentration when the pressurization lasted longer. Thus, the metabolism of microorganisms was significantly inhibited, and the enzyme activity was reduced. It was inferred that a continuous pressure reduced the DHA of bacteria and the antipressure shock performance of the microorganisms [50].

#### 3.3.3. Summary of Pressurized Gas Type Comparison

Detailed index analytical conditions are described in Table 1, including changes in different parameters (SCCP removal rate, bacterial growth, EPS, and DHA) under pure oxygen and air pressurization. The prevention and control of membrane fouling in the membrane treatment process is a large concern, and the main substance that causes membrane fouling is EPS [51]. The results in our investigation (Table 1) indicated that pressurization with pure oxygen mainly led to an increase of TB-EPS in EPS and PN but not PS in LB-EPS, which both had little effect on the fouling of membrane modules. Because some studies have shown that the fouling of membrane modules was mainly caused by the large increase in PS in LB-EPS [52,53], this provides feasibility for the addition of subsequent membrane modules in a pressurized reactor.

### 3.4. Micromorphological Changes of Bacteria under Pressure Conditions

To better explain the experimental phenomenon, SEM images with magnifications of 4500× and 8000× (Figure 5 and Figure 6) were taken to reveal the morphological characteristics of *E. coli* strain 2 under different pressures at 20 mg/L SCCPs. As displayed in Figure 5, the distribution of bacteria growing under atmospheric pressure was loose, and the bacterial surface was smooth. In contrast, the bacterial distribution under pressurization was tight, and the surface became rough and depressed, and filamentous substances surrounded the bacteria, particularly at 0.3 MPa. This might be due to the increased pressure, which caused microorganisms to produce many dense EPS to protect their own cells and tissues, which prevented them from directly contacting the pollutant SCCPs, thereby reducing the internal damage caused by SCCPs [54]. The higher-magnification 8000× images (Figure 6) showed that the individual bacterial cells were the largest and easily distinguishable under atmospheric pressure, while those grown under pressure were not. Additionally, the cells grown under pressure were compressed, resulting in agglomerates and filaments, which aggregated and appeared denser than those grown under atmospheric pressure. Moreover, there were many thin-film tangles on the cell surface after 7 days of culture, and the cell surface appeared to be sticky and uneven. This may have been caused by the accumulation of EPS produced by the strain.

Briefly, adverse conditions or environmental pressure promoted the production of EPS, which is a bacterial self-protection function. EPS provides a complete protective layer for microorganisms to resist severe external conditions, such as toxic organic compounds [18,19] and heavy metals [20], among others. Unfavorable environments may also change the EPS composition, leading to higher levels of cell death and lysis, or even the release of proteins and polysaccharides into bacterial suspensions.

### 3.5. Effects of Different Bacterial Structures on SCCP Removal

#### 3.5.1. Cell Surface Hydrophobicity Analysis

The cell surface hydrophobicity (CSH) of bacteria is a significant parameter that regulates the interactions between bacteria and hydrophobic substrates and other solid surfaces. It also affects the adsorption and degradation of hydrophobic organic matter by bacteria [55]. The changes in the cell surface hydrophobicity of *E. coli* strain 2 before and after EPS extraction are listed in Table 2. The hydrophobicity before extraction was 12.0 ± 0.4%, while the surface hydrophobicity of *E. coli* strain 2 after extraction was 15.2 ± 0.2%. As a typical POP, SCCPs are relatively hydrophobic, with an octanol/water partition coefficient log*K*_ow_ between 4.8 and 7.6 [1]. Therefore, they are easily adsorbed by the hydrophobic region of the bacterial surface. In other words, the increased hydrophobicity of the surface of *E. coli* strain 2 after EPS extraction was more conducive to the adsorption of hydrophobic substances.

#### 3.5.2. Adsorption of SCCPs before and after Bacterial Extraction of EPS

Considering the influence of EPS on the bacterial adsorption of SCCPs, the experiment also extracted the EPS of bacteria and investigated the adsorption removal of SCCPs by LB-EPS, TB-EPS, and *E. coli* strain 2 after complete EPS extraction and *E. coli* strain 2 without EPS extraction. As indicated in Appendix A, the adsorption removal of extracted LB-EPS and TB-EPS on SCCPs was not apparent (<5%). The adsorption effect of *E. coli* strain 2 after complete EPS extraction was approximately 38%, while that of *E. coli* strain 2 without EPS extraction was worse, removing only 35.95% of the SCCPs. This was consistent with the surface hydrophobicity of *E.*
*coli* strain 2 after EPS extraction (15.2 ± 0.2%). This indicated that the EPS had little effect on the adsorption of SCCPs by *E. coli* strain 2, and it was hypothesized that the lipids absorbed most of the SCCPs [56]. Previous observations have revealed that the surface hydrophobicity of bacteria is closely related to the content of surface lipopolysaccharides in the outer bacterial cell walls [57].

Moreover, by extracting all the parts of the bacteria, it was found that LB-EPS and TB-EPS had no obvious adsorption removal effect on SCCPs (both were lower than 5%). The adsorption removal effect of bacteria with EPS (35.95%) was little worse than that of bacteria without EPS (38%).

### 3.6. Possible Degradation Mechanism of Bacterial SCCP Removal

As shown in Figure 7a, when the concentration of SCCPs was 20 mg/L, the peak of SCCPs in a gas chromatogram had obviously decreased after the SCCPs were removed by *E. coli* strain 2. In addition, *E. coli* strain 2 degraded the SCCPs rather than adsorbing them because chloride ions were produced in the reaction solution (Figure 7b).

Furthermore, the GC–MS spectrum is shown in Appendix A. The industrial SCCPs selected for the experiment were mixed pollutants, and the mixture obtained by the GC–MS included medium-chain and long-chain chlorinated paraffins in addition to short-chain chlorinated paraffins. The GC–MS spectrum results show that the mixture in SCCPs was reduced from more than 100 to less than 70, and long chains became short chains after biodegradation. There were no C–Cl bonds in the products, demonstrating that the degradation process of SCCPs may have begun with the rupture of a C–Cl bond [7,13,58], producing Cl^−^. Subsequently, the rupture of a C–C bond occurred [11,13,59]. Since the industrial short-chain chlorinated paraffins were purchased, approximately 10–15% medium- and long-chain chlorinated paraffins were also present. However, after the degradation by *E. coli* strain 2, the GC–MS data show that SCCPs were oxidized and dechlorinated by *E. coli* strain 2 oxidase to generate short-chain alkanes (C_8_ and C_9_), and the three products, 2,4-dimethylheptane (C_9_H_20_), 2,5-dimethylheptane (C_9_H_20_), and 3,3-dimethylhexane (C_8_H_18_) (Appendix A), were determined.

Furthermore, the products obtained under 0.15 MPa and 0 MPa were identical, which implied that pressurization only enhanced the degradation rate of SCCPs by *E. coli* strain 2, without affecting the degradation pathway (data not shown). In general, SCCP congeners with lower chlorination degrees have greater bioavailability to microorganisms, while short-chain alkanes after dechlorination are less toxic [13]. Moreover, there might be other intermediates that could not be detected, possibly because they were unstable or their concentration was lower than the detection limit.

## 4. Conclusions

In the pressurized system, compared with air pressurization, a moderate pure oxygen pressurization was more conducive to the removal of SCCPs by bacteria. A removal rate of 20 mg/L SCCPs by *E. coli* strain 2 increased by 25% at 0.15 MPa. The total amount of EPS increased significantly upon increasing the SCCP concentration and pressure (0–0.3 MPa), and the TB-EPS content was greater than that of LB-EPS. A high pressure mainly promoted the secretion of PN in LB-EPS. The increase in TB-EPS and PN rather than PS in LB-EPS all had little effect on the fouling of membrane modules in the subsequent pressurized membrane reactor. Furthermore, the GC–MS results indicated that the degradation pathway involved the cleavage of the C–Cl bond in SCCPs and the subsequent production of Cl^−^, followed by the cleavage of C–C bonds. In this way, long-chain alkanes were degraded into short-chain alkanes. The main degradation products detected by GC–MS were 2,4-dimethylheptane (C_9_H_20_), 2,5-dimethylheptane (C_9_H_20_), and 3,3-dimethylhexane (C_8_H_18_), regardless of pressurization.

## Figures and Tables

**Figure 1 membranes-12-00634-f001:**
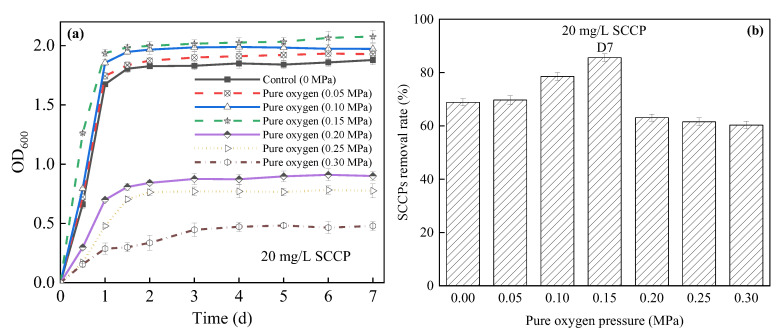
Relationship between different high-purity oxygen pressures and (**a**) the growth of *E. coli* strain 2; (**b**) SCCP removal rate in a pressurized reactor. SCCPs, short-chain chlorinated paraffins.

**Figure 2 membranes-12-00634-f002:**
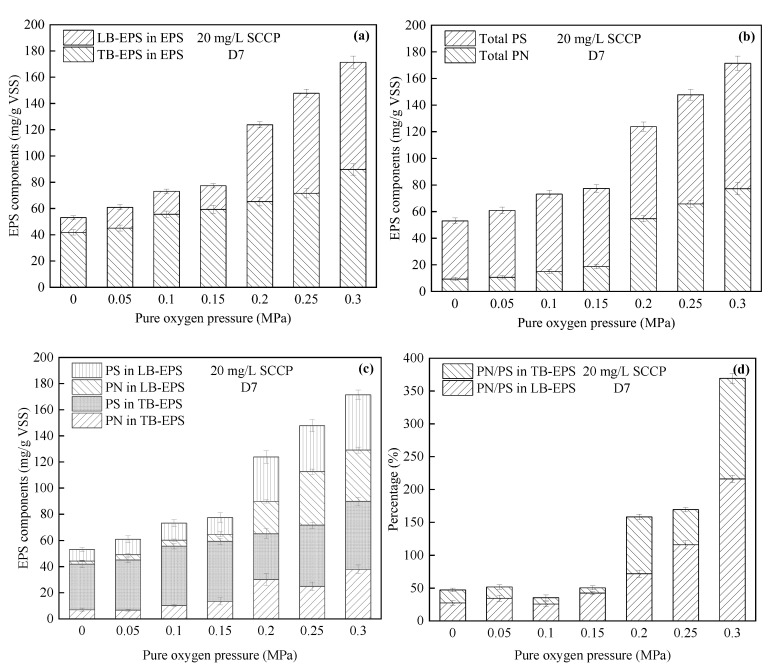
Relationship between different high-purity oxygen pressures and (**a**) LB-EPS and TB-EPS components; (**b**) PS, PN components; (**c**) EPS components; and (**d**) PN/PS ratio of EPS in a pressurized reactor. EPS, extracellular polymeric substances; LB-EPS, loosely bound EPS; PN, protein; PS, polysaccharide; TB-EPS, tightly bound EPS.

**Figure 3 membranes-12-00634-f003:**
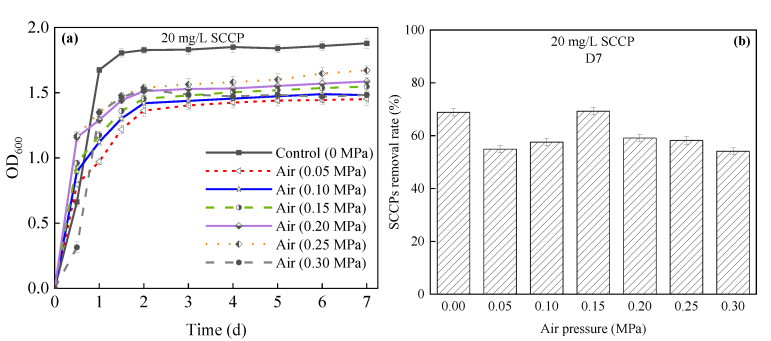
Relationship between different air oxygen pressures and (**a**) the growth of *E. coli* strain 2; (**b**) SCCP removal rate in a pressurized reactor. SCCPs, short-chain chlorinated paraffins.

**Figure 4 membranes-12-00634-f004:**
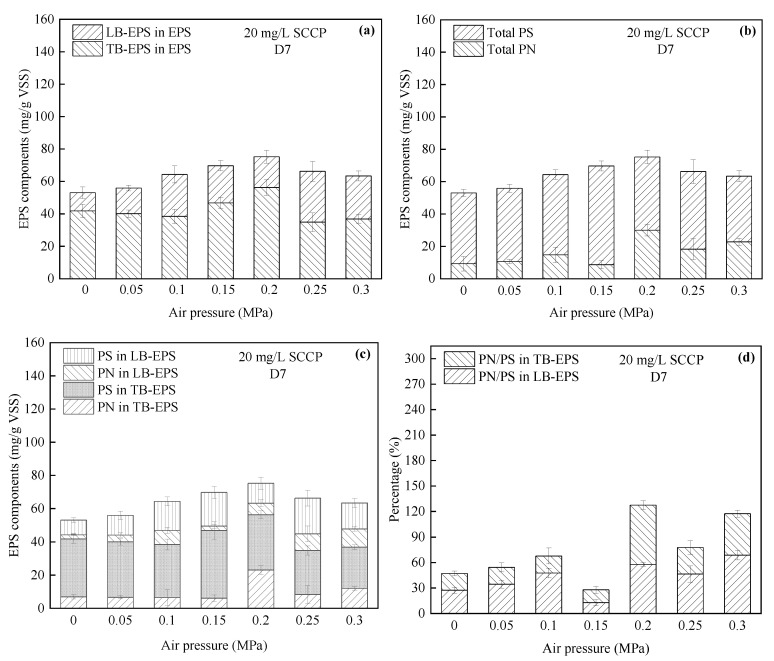
Relationship between different air pressures and (**a**) LB-EPS and TB-EPS components; (**b**) PS and PN components; (**c**) EPS components; and (**d**) PN/PS ratio of EPS in a pressurized reactor. EPS, extracellular polymeric substances; PN, proteins; PS, polysaccharides.

**Figure 5 membranes-12-00634-f005:**
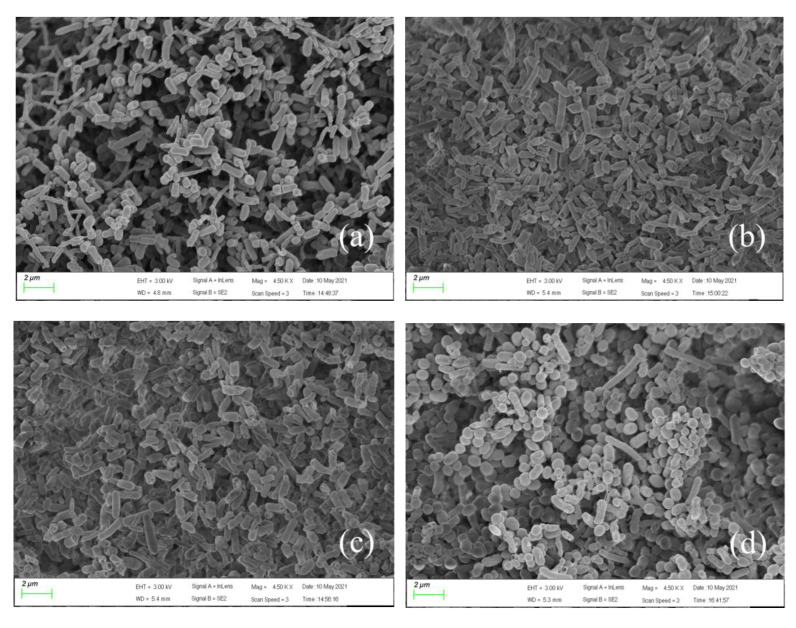
SEM images of *E. coli* strain 2 under different pressures: (**a**) 0 MPa, (**b**) 0.1 MPa, (**c**) 0.15 MPa, (**d**) 0.2 MPa, and (**e**) 0.3 MPa (the magnification is 4500×; the scale bar = 2 μm). SEM, scanning electron microscopy.

**Figure 6 membranes-12-00634-f006:**
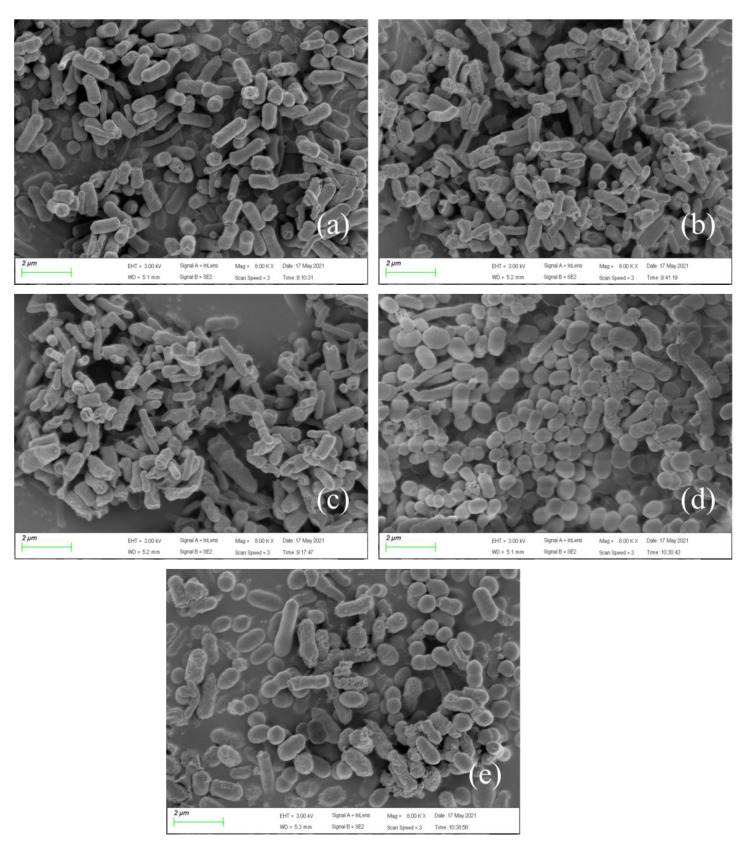
SEM images of *E. coli* strain 2 under different pressures: (**a**) 0 MPa, (**b**) 0.1 MPa, (**c**) 0.15 MPa, (**d**) 0.2 MPa, and (**e**) 0.3 MPa (the magnification is 8000×; the scale bar = 2 μm). SEM, scanning electron microscopy.

**Figure 7 membranes-12-00634-f007:**
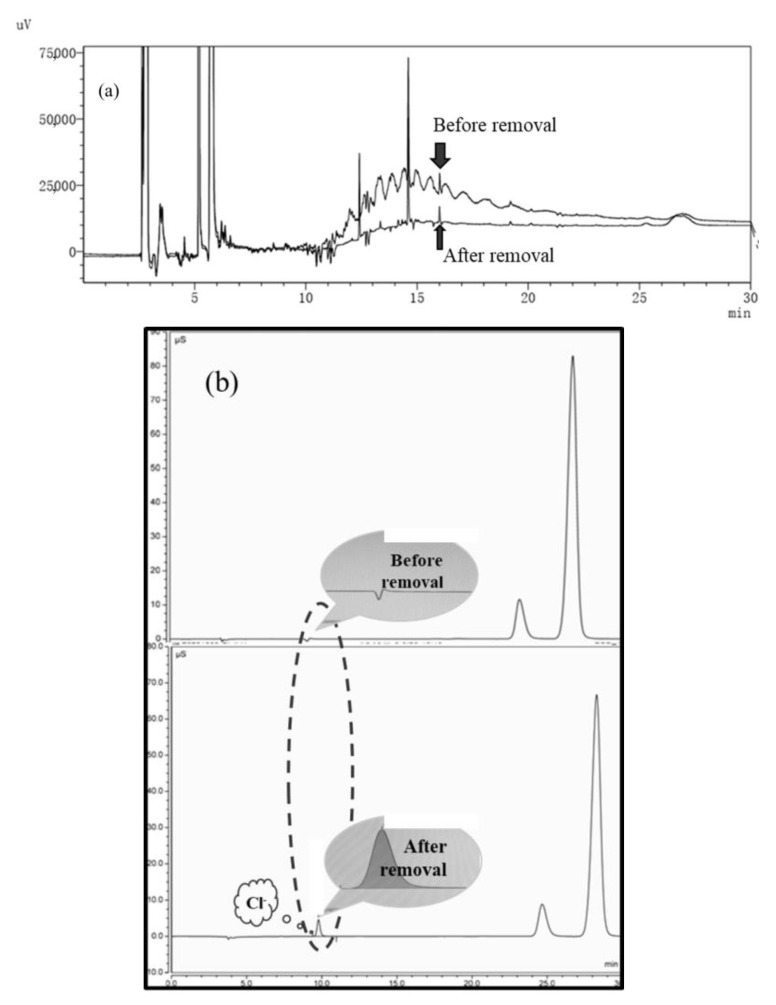
*E. coli* strain 2 degradation of SCCPs: (**a**) changes in the SCCPs; (**b**) changes in the concentration of chloride ions in solution. SCCPs, short-chain chlorinated paraffins.

**Table 1 membranes-12-00634-t001:** Comparison of the parameters of pure oxygen and air pressurization.

Parameter	Pressurization (0–0.3 MPa)	
	Pure Oxygen Pressurization	Air Pressurization
SCCP removal rate	The maximum removal rate was 85.61% (0.15 MPa), which was greater than 0.05 and 0.1 MPa, while the removal rate decreased (>0.15 MPa).	The removal rate reached the highest value of 69.28% (0.15 MPa), and the removal rate decreased (>0.15 MPa).
OD_600_	In the range of 0.05–0.15 MPa, the pressure did not affect the growth of microorganisms, but it was severely inhibited (>0.15 MPa).	Pressure did not promote bacterial growth (compared with 0 MPa).
EPS	Upon increasing the pressure, the EPS content increased significantly, among which TB-EPS was the main EPS type. A high pressure promoted the secretion of PN in LB-EPS.	The EPS content under pressurization increased compared with EPS under atmospheric pressure. TB-EPS accounted for more than 50% of the total EPS, and more PS was always generated than PN.
DHA	The DHA was promoted by an appropriate low pressure but inhibited by a high pressure (>0.15 MPa).	The DHA was lower than at atmospheric pressure.

Note: SCCPs, short-chain chlorinated paraffins; EPS, extracellular polymeric substances; PN, proteins; PS, polysaccharides; DHA, dehydrogenase.

**Table 2 membranes-12-00634-t002:** Cell surface hydrophobicity for *E. coli* strain 2 cell surface before and after EPS extraction.

Sample	Cell Surface Hydrophobicity of Bacteria
*E. coli* strain 2 before extraction of EPS	12.0 ± 0.4%
*E. coli* strain 2 after extraction of EPS	15.2 ± 0.2%

## Data Availability

The data presented in this study are available on request from the corresponding author.

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
