# Peer review of "Effects of Pressurized Aeration on the Biodegradation of Short-Chain Chlorinated Paraffins by Escherichia coli Strain 2"

_membranes, 2022, doi:10.3390/membranes12060634_

Round 1
Reviewer 1 Report
This article does not contain any new findings that are not mentioned in the cited literature. The article is clearly written and clear, tables and graphs are correctly described. I would recommend to the authors to emphasize more what new this article brings.
Reviewer 2 Report
The manuscript entitled "Effects of pressurized aeration on the biodegradation of short chain chlorinated paraffins by Escherichia coli strain 2" by Qian et al on biodegradation of persistent organic pollutant SCCP by an E. coli strain. Biodegradation of pollutants is a very relevant topic, however, the present manuscript lacks as many as the following points, making it unsuitable for publication at this version, and needs significant revisions:
1. The manuscript looks to have forced languages; many words do not create any sense; for example, Line no. 19 "domesticated bacteria", Line no. 139 "until later use", and more.
2. The English language has major singular/plural issues; is/are, was/were, has/have.
3. Why the rotation speed was particularly chosen from 320 rpm is not cited.
4. The "Materials and Methods" section does not exactly correspond to the "Results and Discussion" section. Many methodologies are missed cited. Both must have corresponding headings so that it will be easy for the readers to understand the work.
5. Figure 1 is not adding much information to the experiment and may be transferred to supplementary information.
6. The magnification at 4500× for all the 5 pressures do not comply with image integrity. The authors are suggested to provide clear images without any modifications.
7. In order to link EPS and high pressure tolerance, it is important to show EPS networks via microscopy.
8. The GC-MS images need to be supplemented.
9. Overall, more references are needed for the present year and the last 2 years to keep the work up to date.
Reviewer 3 Report
Membranes Manuscript ID: membranes-1744516:
The manuscript entitled “Effects of pressurized aeration on the biodegradation of short-chain chlorinated paraffins by Escherichia coli strain 2” by Yongxing Qian et al, summarizes the newest information about a pressurized reactor was introduced, and the removal performance of SCCPs by screened and domesticated bacteria Escherichia coli strain 2 was investigated.
This manuscript describes the existing data in context to that moderate pure oxygen pressurization promoted bacterial growth, but when it exceeded 0.15 MPa, the bacterial growth was severely inhibited. Moreover, the main detected degradation products were 2,4-dimethylheptane (C9H20), 2,5-dimethylheptane (C9H20), and 3,3-dimethylhexane (C8H18).
The authors have reported introduced new pressurized Reactor in the manuscript, but there are certain sections that need to be elaborated and some needs the addition of literature.
General comments:
Ø Authors have mentioned and concluded that in the pressurized system, compared with air pressurization, a moderate pure oxy-514 gen pressurization was more conducive to the removal of SCCPs by bacteria. The removal 515 rate of 20 mg/L SCCPs by E. coli strain 2 increased by 25% at 0.15 MPa, but I wonder whether this system would work in specific conditions or there is no pre-conditioning required for this system to work.
Ø Also, I wonder whether these studies are enough to conclude that this is the ideal system as compared to the current SCCPs removal methods
Round 2
Reviewer 1 Report
I recommend the article to the press.
Author Response
The authors would like to thank the reviewer for his or her suggestion.
Reviewer 2 Report
The revision looks quite impressive. However, small suggestions are given to the authors so that the manuscript becomes publishable:
1. Please add the GC-MS spectrum, gas, and ion chromatogram to the main manuscript and the subsequent materials/ methods and results/ discussions too.
2. The red circles in the SEM images need to be either mentioned/ explained what does they mean, or it is better to remove those circles.
Author Response
Point 1: Please add the GC-MS spectrum, gas, and ion chromatogram to the main manuscript and the subsequent materials/ methods and results/ discussions too.
Response 1: Thank you very much.We revised the paper as required.
Point 2: The red circles in the SEM images need to be either mentioned/ explained what does they mean, or it is better to remove those circles.
Response 2: Thank you very much. This is a good advice. We deleted the red circles as required for these circles, since they were meaningless.
Reviewer 3 Report
Membranes Manuscript ID: membranes- 1744516:
The Revised manuscript entitled “Effects of pressurized aeration on the biodegradation of short-chain chlorinated paraffins by Escherichia coli strain 2” by Yongxing Qian et al, summarizes the significant information about a pressurized reactor was introduced, and the removal performance of SCCPs by screened and domesticated bacteria Escherichia coli strain 2 was investigated.
This manuscript describes the existing data in context to that moderate pure oxygen pressurization promoted bacterial growth, but when it exceeded 0.15 MPa, the bacterial growth was severely inhibited. Moreover, the main detected degradation products were 2,4-dimethylheptane (C9H20), 2,5-dimethylheptane (C9H20), and 3,3-dimethylhexane (C8H18).
The authors have revised the manuscript.
General comments:
Ø Authors should properly look for any grammatical mistakes, along with typo’s.
Ø Figure 5 and 6 should be analyzed and quantified, like any scatter plots should me made properly.
Ø Plagiarism should be checked properly
Author Response
The reviewer's response is in the attachment.

Round 3
Reviewer 3 Report
The Revised manuscript entitled “Effects of pressurized aeration on the biodegradation of short-chain chlorinated paraffins by Escherichia coli strain 2” by Yongxing Qian et al, summarizes the significant information about a pressurized reactor was introduced, and the removal performance of SCCPs by screened and domesticated bacteria Escherichia coli strain 2 was investigated.
Authors have revised the manucript properly